# Microencapsulated Sodium Butyrate Alleviates Immune Injury and Intestinal Problems Caused by Clostridium Perfringens through Gut Microbiota

**DOI:** 10.3390/ani13243784

**Published:** 2023-12-08

**Authors:** Ting Yang, Yaowei Sun, Zhenglie Dai, Jinsong Liu, Shiping Xiao, Yulan Liu, Xiuxi Wang, Shenglan Yang, Ruiqiang Zhang, Caimei Yang, Bing Dai

**Affiliations:** 1College of Animal Science and Technology, College of Veterinary Medicine, Key Laboratory of Applied Technology on Green-Eco-Healthy Animal Husbandry of Zhejiang Province, Zhejiang Provincial Engineering Laboratory for Animal Health Inspection and Internet Technology, Zhejiang Agricultural and Forestry University, Hangzhou 311300, China; yangtingly@126.com (T.Y.); 19550177705@163.com (Y.S.); mmddaa0323@163.com (Z.D.); 15261825191@163.com (X.W.); yangshenglan7@163.com (S.Y.); zrq1034@163.com (R.Z.); 2Zhejiang Vegamax Biotechnology Co., Ltd., Huzhou 313300, China; huijialjs@163.com (J.L.); xiao1974920@126.com (S.X.); yulanflower@126.com (Y.L.); yangcaimei2012@163.com (C.Y.)

**Keywords:** microencapsulated sodium butyrate, broiler chicken, immunity, cecum microflora, *Clostridium perfringens*

## Abstract

**Simple Summary:**

Necrotic enteritis is an enterotoxemic disease caused by *Clostridium perfringens*, leading to diarrhea or necrotizing lesions in the intestines of animals, with severe cases leading to death. The butyrate attenuated the inflammatory response and improved intestinal health in piglets challenged with pathogenic bacteria, but it was absorbed by the anterior segment of the gastrointestinal tract. Microencapsulation is a simple and effective method to prevent butyrate from absorption. Thus, we evaluate the effectiveness to two butyrate alleviates clostridium perfringens infections. Our results indicate that dietary supplementation with sodium butyrate or microencapsulated sodium butyrate improves the immune status and morphology of intestinal villi, increases the production of VFAs, and modulates cecal microbiota in chickens challenged with *Clostridium perfringens*. Moreover, microencapsulated sodium butyrate contains less butyrate than sodium butyrate. These findings indicate that microencapsulated sodium butyrate was more effective than sodium butyrate with the same butyrate supplemental amount.

**Abstract:**

Microencapsulated sodium butyrate (MS-SB) is an effective sodium butyrate additive which can reduce the release of sodium butyrate (SB) in the fore gastrointestinal tract. In this study, we assess the protective effects and mechanisms of MS-SB in *Clostridium perfringens* (*C. perfringens*)-challenged broilers. Broiler chickens were pre-treated with SB or MS-SB for 56 days and then challenged with *C. perfringens* three times. Our results indicate that the addition of MS-SB or SB before *C. perfringens* infection significantly decreased the thymus index (*p* < 0.05). Serum IgA, IgY, and IgM concentrations were significantly increased (*p* < 0.05), while pro-inflammatory IL-1β, IL-6, and TNF-α were significantly decreased (*p* < 0.05) under MS-SB or SB supplementation. Compared with SB, MS-SB presented a stronger performance, with higher IgA content, as well as a lower IL-1β level when normal or C. perfringens-challenged. While *C. perfringens* challenge significantly decreased the villus height (*p* < 0.05), MS-SB or SB administration significantly increased the villus height and villus height/crypt depth (V/C ratio) (*p* < 0.05). Varying degrees of SB or MS-SB increased the concentrations of volatile fatty acids (VFAs) during *C. perfringens* challenge, where MS-SB presented a stronger performance, as evidenced by the higher content of isovaleric acid and valeric acid. Microbial analysis demonstrated that both SB or MS-SB addition and *C. perfringens* infection increase variation in the microbiota community. The results also indicate that the proportions of *Bacteroides*, *Faecalibacterium*, *Clostridia*, *Ruminococcaceae*, *Alistipes*, and *Clostridia* were significantly higher in the MS-SB addition group while, at same time, *C. perfringens* infection increased the abundance of *Bacteroides* and *Alistipes*. In summary, dietary supplementation with SB or MS-SB improves the immune status and morphology of intestinal villi, increases the production of VFAs, and modulates cecal microbiota in chickens challenged with *C. perfringens*. Moreover, MS-SB was more effective than SB with the same supplemental amount.

## 1. Introduction

Antibiotic growth promoters have been used to improve animal growth and to control or prevent animal disease during rearing; however, many countries have forbidden the dietary use of antimicrobial agents to avoid the emergence of antibiotic-resistant bacteria [1]. Thus, animal nutritionists have been attempting to discover antibiotic alternatives that are low-cost, widely used, and which have an obvious effect. Butyric acid has a positive effect on the normal intestinal mucosa, which can be attributed to a major energy source for intestinal epithelial cells [2,3]. In the intestine, an appropriate concentration of butyric acid is beneficial in protecting against the invasion of micro-organisms, which may be attributed to the effects of VFAs in terms of maintaining the pH of the intestinal lumen [4]. In a previous study, butyrate significantly ameliorated the mice intestine and intestinal epithelial cell inflammatory response and intestinal epithelium barrier dysfunction caused by 2,4,6-trinitrobenzene sulphonic acid (TNBS) [5]. Butyrate also has an immunomodulatory effect; for example, butyrate attenuated steatohepatitis through restoring the dysbiosis of gut microbiota. As such, butyrate has been considered as a potential gut microbiota modulator and therapeutic substance for non-alcoholic fatty liver disease (NAFLD) [6].

In the context of animal rearing, the harsh environment, stress, harmful bacteria, and other factors may result in intestinal health damage, expressed as intestinal inflammatory responses, intestinal barrier injury, and intestinal flora imbalances. Necrotic enteritis is an enterotoxemic disease caused by *C. perfringens*, leading to diarrhea or necrotizing lesions in the intestines of livestock and humans and, in severe cases, leading to death, with mortality rates of up to 100% in affected piglets [7]. As one of the most economically important diseases affecting poultry worldwide, necrotic enteritis causes $6 billion in annual losses globally [8]. In animals, associated lesions caused by *C. perfringens* have been found in the whole intestine, and were significantly more severe in the jejunum compared to the duodenum and ileum [9]. In healthy chickens, it is hard to find *C. perfringens* spores in the gastrointestinal tract, but the total number of vegetative *C. perfringens* cells increased when necrotic enteritis occurred, presenting a positive correlation between presence in the duodenum, jejunum, and ileum and disease severity [9,10]. In a retrospective study, *C. perfringens* was also found to cause liver morphological lesions, necrotizing hepatitis, congested lungs, and neurological diseases, manifesting as tremors, stargazing, and incoordination [11].

In a previous study, dietary supplementation with butyrate attenuated the inflammatory response and improved intestinal health in piglets challenged with enterotoxigenic Escherichia coli (ETEC) through inhibiting the activation of NF-κB/MAPK and modulating the hindgut microbiota [12]. Another study used an adherent-invasive Escherichia coli (AIEC) challenge model and showed that butyrate augmented AIEC invasiveness, while concurrently bolstering the intestinal epithelial barrier and reducing intestinal inflammation [13]. However, to the best of our knowledge, there have been few studies on the effect of butyrate in mitigating *C. perfringens* infection in broilers. In addition, butyrate given orally is quickly absorbed and used as energy by mucosal cells, absorbed and metabolized by the bird ingluvies and throughout the whole gastrointestinal tract, limiting the amount of butyrate that reaches the hindgut and restricting its practical use in the animal production context [14]. In order to prevent butyrate from absorption and metabolization in the anterior segment of the gastrointestinal tract, encapsulation is a simple and effective method. Previous studies have shown that microencapsulation reduces the release of contents into gastrointestinal fluid [15].

Here, we propose the hypothesis that microencapsulated sodium butyrate has a better protective effect than sodium butyrate in broilers challenged with *C. perfringens*. Consequently, we systematically assess the protective and underlying mechanisms of sodium butyrate (SB) and microencapsulated sodium butyrate (MS-SB) against *C. perfringens* infection in broilers. This study possesses considerable theoretical significance in relation to protection against the occurrence of necrotic enteritis and providing a strategy for effectively utilizing gastrointestinal-sensitive biological agents.

## 2. Materials and Methods

### 2.1. Animals and Diets

A total of 360 1-day-old male yellow feather broiler chickens were obtained from a commercial hatchery. The chickens were completely randomly allocated into 6 groups, with 6 replicates, and each replicate containing 10 chickens. The study adopted a completely randomized design with a 3 × 2 factorial pattern (3 kinds of diet and *C. perfringens*-challenged or not). The chickens were reared in a temperature-controlled room and maintained on a 24 h constant light schedule, and allowed ad libitum access to feed and water. The experiment last 57 days. The basal diet was formulated based on the NRC (1994) [16] and nutrient requirements for yellow chickens [17]; see Table 1. During the experimental period, chickens were fed a basal diet (basic), a basal diet with 1000 mg/kg sodium butyrate (SB), or 1000 mg/kg microencapsulated sodium butyrate (MS-SB), respectively. At 53 days old, the challenged group of chickens were challenged with 1 mL *C. perfringens* suspension (10^9^ CFU/mL) via intragastric administration every other day, and the non-challenged group of chickens were given 1 mL media in the same way. The *C. perfringens* used in this study were kept in our laboratory. The SB contained 98% sodium butyrate, and was purchase from Wuhan Jiyesheng Chemical Co., Ltd. (Wuhan, China). The MS-SB was coated with a polymer enteral material and contained 40% sodium butyrate, and was provided by Zhejiang Vegamax Biotechnology Co., Ltd. (Huzhou, China).

### 2.2. Sample Collection

At 58 days old, blood was collected from the jugular vein after the chickens were starved overnight and weighted. It was centrifuged at 3500× *g* for 15 min at 4 °C, then stored at −20 °C for future analyses. The chickens were slaughtered by cervical dislocation. The middle segments of jejunum were collected and fixed with 4% paraformaldehyde. The cecal contents were aseptically collected and assessed for volatile fatty acids (VFAs) and microflora composition.

### 2.3. Organ Index

The liver, spleen, thymus, and bursa of Fabricius of each sampling broiler were removed, the blood stains were wiped off the surface, then they were weighed. The relative organ weights were calculated as per the following equation:Organ weight indexes = Organ weight (g)/Body weight (g) × 100

### 2.4. Serum Immune Indicators

The concentrations of serum immunoglobulin A (IgA) (CAS:ANG-E32004C; 10–600 ng/mL), immunoglobulin M (IgM) (CAS:ANG-E32005C; 0.1405–11.25 μg/mL), immunoglobulin Y (IgY) (CAS:ANG-E32209C; 0.062.5–3.75 ng/mL), interleukin-1β (IL-1β) (CAS:ANG-E32031C; 1.875–112.5 ng/L), interleukin-6 (IL-6) (CAS:ANG-E32013C; 1–60 ng/L), and tumor necrosis factor-α (TNF-α) (CAS:ANG-E32030C; 1.25–75 ng/L) were measured using an enzymatic chromatometric method using the ELISA Kits that were purchased from Angle Gene Biotechnology Co., Ltd. (Nanjing, Jiangsu, China), according to the manufacturer’s instructions.

### 2.5. Jejunum Morphology Analysis

The fixed and pruned jejunum was dehydrated with gradient ethanol, followed by cleaning with xylene, waxing, embedding, slicing, and staining with hematoxylin–eosin (HE). Finally, we took pictures using a microscope (Nikon, Tokyo, Japan). Ten intact villi for each sample were randomly selected to measure the villus height (V) and crypt depth (C) using the Image Pro Plus 6.0 software (Rockville, MD, USA), and the ratio of villus height to crypt depth (V/C) was calculated.

### 2.6. Volatile Fatty Acid (VFA) Analysis

The VFAs in cecum content were determined via gas chromatography according to the method of Yu et al. (2023) [18]. In brief, a sample containing about 0.5 g of cecum content was mixed with pre-cooled water at a mass volume ratio of 1:3 and centrifuged at 12,000× *g* for 10 min at 4 °C. The supernatant was mixed with 25% metaphosphoric acid in a 5:1 ratio and rested for 30 min, followed by centrifugation at 10,000× *g* for 10 min at 4 °C. Then, the supernatant was transferred into a sample bottle for testing using an Agilent Technologies 7890B GC System (column parameter: 30 m × 0.25 mm × 0.25 μm; Agilent Technologies, Santa Clara, CA, USA). Pure acetic acid, propionic acid, isobutyric acid, butyric acid, isovaleric acid, and valeric acid solutions were used as standards to calculate relevant concentrations in the sample.

### 2.7. Cecum Microflora

Total bacterial DNA of cecum microbiota was extracted using a DNA extraction kit, and the DNA concentration was determined using agarose gel electrophoresis. The V3–V4 regions of 16S rRNA were amplified with universal primers 515F/806R using the Applied Biosystems GENEAMP 9700 (Thermo Fischer Scientific, Waltham, MA, USA). The amplification products were recovered after purification using an AXYGEN DNA Gel Extraction Kit (Union City, CA, USA). The amplification products were sequenced on an Illumina miseq (PE300) platform provided by Majorbio Co., Ltd. (Shanghai, China), after quantitative analysis conducted using a quantifluortm blue fluorescence quantitative system (Promega, WI, USA). In order to obtain the Amplicon Sequence Variant (ASV) variants and feature lists, the DADA2 variants in QIIME2 were used to optimize the data obtained through sequencing. The alpha diversity component, Shannon, Chao, and Simpson indices were adopted to indicate the diversity of microbiota. Principal component analysis (PCA) and principal co-ordinates analysis (PCoA) were conducted to analyze the species indices, including the β-diversity component between different groups. Finally, linear discriminant analysis coupled with effect size (LEfSe) was used to identify microbial differences among all treatment groups.

### 2.8. Statistical Analysis

The data were tested for normality and homogeneity of variance through the Levene test. Then, two-factor analysis of variance and Tukey’s HSD were carried out using the JMP Pro software 13.0 (SAS, Carrey, MS, USA). The model equation included the main effects (sodium butyrate addition and *C. perfringens* challenge) and their interactions. The differences among treatments were tested using Tukey’s test when the main effects or interactions were significant. The statistical significance was set to *p* < 0.05. All values are shown as mean ± SEM. The resulting data were plotted using the GraphPad Prism 8.0 software (GraphPad Software, San Diego, CA, USA).

## 3. Results

### 3.1. Microencapsulated Sodium Butyrate Alleviated C. perfringens Infection

At 53–57 days old, the broilers were challenged with *C. perfringens* three times through gavage administration every other day. As shown in Table 2, the *C. perfringens* challenge significantly decreased the thymus index and significantly increased the spleen index (*p* < 0.05). The addition of SB significantly decreased the thymus index (*p* < 0.05), and MS-SB had the same effect as sodium butyrate. It is worth noting that the addition of SB and MS-SB decreased the thymus index under *C. perfringens* infection.

### 3.2. Microencapsulated Sodium Butyrate Alleviated Reduced Systemic Inflammation Caused by C. perfringens

The levels of inflammatory factors and immunoglobulin in sera are shown in Figure 1. The *C. perfringens* challenge had no influence on inflammatory factor and immunoglobulin content in sera. However, SB addition significantly increased the immunoglobulin content in sera, including IgA, IgM, and IgY (*p* < 0.05). In addition, SB significantly decreased the content of IL-1β, IL-6, and TNF-α (*p* < 0.05). Compared with SB, MS-SB presented a stronger performance, with higher IgA content, as well as a lower IL-1β level when normal or *C. perfringens*-challenged.

### 3.3. Microencapsulated Sodium Butyrate Repaired Intestinal Morphology Damaged by C. perfringens

We also measured the parameters of jejunum villi through HE stains, and the results are shown in Figure 2. The results revealed that SB administration significantly increased the villus height and V/C ratio (*p* < 0.05), especially the MS-SB addition. However, *C. perfringens* challenge significantly decreased the villus height (*p* < 0.05) without affecting the V/C ratio.

### 3.4. Microencapsulated Sodium Butyrate Ameliorated VFAs under C. perfringens Challenge

For this study, we also detected the VFA content in the broiler cecum, and the results are shown in Figure 3. The results indicated that the *C. perfringens* challenge had no effect on the VFA content. However, the addition of SB significantly decreased the content of acetic acid (*p* < 0.05) under *C. perfringens* non-challenge; at same time, SB or MS-SB presented varying degrees of increased fatty acid concentrations during *C. perfringens* challenge. It should be noted that MS-SB had a stronger performance, with a higher content of isobutyric acid, isovaleric acid, and valeric acid.

### 3.5. Microencapsulated Sodium Butyrate Modulated Gut Microbiota Community Variation Caused by C. perfringens

The changes in the diversity of cecum microbiota are summarized in Figure 4. A total of 921 ASVs were shared among the five treatment groups based on the Venn diagram, with non-overlapping regions indicating unique OTUs in the CON group (*n* = 569), SB group (*n* = 658), MS-SB group (*n* = 743), CON-CP group (*n* = 592), and MS-SB-CP group (*n* = 611); see Figure 4A. We adopted the Chao index, Shannon index, and Simpson index to assess the microbiota community diversity. The results indicated that *C. perfringens* infection and SB addition individually presented more significant diversity in cecum microbiota when compared to the CON group, but there was no difference between them (Figure 4B–D). Principal component analysis (PCA) and principal coordinate analysis (PCoA) were performed to investigate the differences in species complexity and structural alterations in microbial communities. PCoA and PCA plots revealed the degree of diversity discrepancy in cecum microbiota between different treatments; in particular, the PCoA plot indicated that the CON group was separated from the other groups (Figure 4E,F).

### 3.6. Microencapsulated Sodium Butyrate Modulated Gut Microbiota Community Composition Caused by C. perfringens

In order to explore the change in cecum microbiota structure, we analyzed the relative abundance of cecum microbiota at phylum to species levels. At the phylum level, about 4 major phyla were detected, including *Firmicutes*, *Bacteroidota*, *Verrucomicrobiota*, and *Actinobacteriota* (Figure 5A), and the relative abundance of *Proteobacteria* was higher in the *C. perfringens* infection groups than non-infection groups. The addition of MS-SB significantly increased the relative abundance of *Campilobacteria* (Figure 5D). At the genus level, about 13 major phyla were detected as dominant genera (Figure 5B). The relative abundances of *Bacteroides* in the *C. perfringens* infection groups were higher than in the non-infection groups, and the relative abundances of *Ruminococcus_torques* and *Alistipes* in the MS-SB-CP groups were higher than in other groups (Figure 5E). At the species level, about 11 major phyla were detected as dominant flora (Figure 5C). In addition, the relative abundances of 6 bacteria were higher with MS-SB addition than in the CON group, including *Bacteroides*, *Faecalibacterium*, *Clostridia_UCG-014*, *Ruminococcaceae*, *Alistipes_inops*, and *Clostridia_vadinBB60*. The *C. perfringens* infection increased the abundance of *Bacteroides* and *Alistipes_inops* (Figure 5F). Finally, we analyzed the association between VFAs and microbiota composition. The results demonstrated that *Ruminococcaceae_torques* and *Alistipes* were positively correlated with isobutyric acid concentration, while *Lachnospiraceae*, *Ruminococcaceae_torques*, and *Alistipes* were positively correlated with valeric acid (Figure 6A). At the species level, *Ruminococcaceae_torques* and *Alistipes* were positively correlated with isobutyric acid, and *Lachnospiraceae*, *Ruminococcaceae_torques*, and *Alistipes* were positively correlated with valeric acid (Figure 6B).

## 4. Discussion

In intensive animal production, the animals are faced with a variety of external factors at any time, such as production environment stress and physiological stress. In healthy animals, bacteria and health stand at either end of a pair of scales; once this balance is broken, harmful bacteria in the environment and body can cause damage to animal health, including intestinal inflammation, immune stress, and so on [19]. *C. perfringens* is consumed by chickens from environmental sources during rearing, including contaminated feed, water, and the farm environment [20]. As previously reported, the immune organ changes in response to infection with *C. perfringens*, shown in terms of the morphology and weight of the bursa of Fabricius, spleen, and thymus [21]. In this study, the results indicated that the spleen index increased, which is consistent with previous findings, as well as showing expected changes in the thymus index. This may be attributed to the age and species, as broilers near the end of growth were used in this study. Thus, these results indicate the validity of the *C. perfringens* infection model.

Moreover, we demonstrated that the addition of SB or MS-SB had no promotive or protective effects on the immune organs in this study. The results are not consistent with the results of a previous study, in which the thymus, spleen, and bursa weighed more in the SB addition group compared with control group [22]. Moreover, in a study in quails, dietary supplementation with 1000 mg/kg SB significantly increased the thymus and bursa of Fabricius [23]. As a matter of fact, we have also confirmed that SB and MS-SB could enhance immune organ development throughout the growth period in a previous study (unpublished). Thus, according to the results of this study, SB or MS-SB have no protective effect against the immune organ changes caused by *C. perfringens* infection, evidenced by the fact that SB or MS-SB were unable to reverse the organ changes induced by *C. perfringens*.

The immune organs, immune cells, and immune molecules make up the immune system of animals, and can be categorized into two parts—the innate immune system and the adaptive immune system—according to the manner in which they act against invading pathogens [24]. As our results indicated that SB and MS-SB could enhance immune organ development during the growth period (unpublished), we further determined the levels of immunoglobulins and immunomodulatory cytokines in sera. Immunoglobulins are synthesized and secreted by B-cells after immune cells are activated by antigens, which can bind to specific antigens in order to defend against invading pathogens [25]. In this study, the results showed that *C. perfringens* infection decreased the immunoglobulin levels in sera, while SB and MS-SB improved the secretion of immunoglobulins, consistent with the results of previous studies [26]. However, it should be noted that the enhancement brought by SB and MS-SB was diminished under *C. perfringens* infection, when compared to normal conditions. Immunomodulatory cytokines are produced by immune cells and act on other immune cells, which can be classified as pro-inflammatory or anti-inflammatory according to their function [27]. Pro-inflammatory cytokines (e.g., IL-1α/β, TNF-α/β, and IL-6) up-regulate inflammatory reactions, while anti-inflammatory cytokines (e.g., IL-10) down-regulate inflammatory responses and promote tissue healing [28]. In rat models, necrotic enteritis causes enteric inflammation accompanied by serum proinflammatory cytokine production [29]. Thus, we proceeded to determine the content of inflammatory factors in sera. The results demonstrated that SB or MS-SB exhibited a potent anti-inflammatory effect, evidenced by reductions in IL-1β, IL-6, and TNF-α, especially in the case of the slight increase in inflammatory factors caused by *C. perfringens*. Similar to the results of our study, the study published by Sun et al. (2021) reported that sodium butyrate inhibited intestinal inflammation through the HMGB1-TLR4/NF-κB pathway [3]. Therefore, we speculate that SB or MS-SB may enhance immune function by regulating serum inflammatory cytokines in broiler chickens. Moreover, compared with SB, MS-SB presented a better anti-inflammatory effect.

As an organ for nutrient digestion and absorption, the intestinal tract is also an important barrier to maintain the homeostasis of the internal environment, and is the first barrier to deal with foreign harmful bacteria [30]. Normal intestinal permeability prevents water and electrolyte loss, promotes the absorption of dietary nutrients, and prevents the entry of antigens and micro-organisms into the body, which are dependent on the integrity of the intestinal villi [31]. In a previous study, necrotic enteritis caused by *C. perfringens* was mostly observed in the jejunum, manifesting as intestinal morphological damage and inflammation [9]. In addition, SCFA can improve the proliferation of gut epithelial cells and increase their villi height, which subsequently helps to improve the capacity of the intestine for nutrient absorption [32]. Thus, we detected the morphology of the jejunum using HE staining, and the results indicated that *C. perfringens* negatively affected the intestinal villi morphology, decreasing both the villus height and crypt depth. These results are similar to those previously reported in broilers [33,34] and mice [35]. We cannot ignore that the addition of SB or MS-SB alleviated the morphological damage to the intestinal villi, and the performance of MS-SB was particularly prominent compared with that of SB. Similar results have also been confirmed in a study using a mouse model [5].

In previous research, VFAs have been shown to inhibit pathogenic micro-organisms and increase the absorption of nutrients, which may contribute to a reduction in the luminal pH [36]. Acetic acid is the shortest fatty acid with a carbon chain, which has been shown to be an intermediary involved in *bifidobacteria* inhibiting the proliferation of intestinal pathogens [37]. Propionate and butyrate have been reported to assist in controlling intestinal inflammation by inducing the differentiation of T-regulatory cells, and the inhibition of histone deacetylation may be involved in the relevant regulatory mechanism [2,38]. In addition, recent studies have shown that valerate inhibits the proliferation of *Clostridium difficile* in the intestinal tract, thereby protecting or treating intestinal diseases [39]. Moreover, butyric acid is usually produced in the large intestine by intestinal bacteria and plays important roles, such as fueling intestinal epithelial cells and increasing mucin production, which may result in changes in bacterial adhesion and improved tight-junction integrity [40,41]. Thus, VFAs seem to play an important role in the maintenance of gut barrier function [42]. We also measured the concentrations of VFAs in the cecum, such as acetic acid, butyric acid (isobutyric acid), and valeric acid (isovaleric acid). The results indicated that *C. perfringens* infection has no effect on the VFA content in the cecum. However, SB or MS-SB increased the concentrations of fatty acids to varying degrees under the *C. perfringens* challenge. Our results are partially consistent with previous findings in broilers [43], and the dietary composition and butyric acid originating from foregut may be the cause of this result, as a result of butyric acid and other VFAs being produced through the bacterial fermentation of unabsorbed carbohydrates or food scraps [44]. Meanwhile, MS-SB was used in our study, which can significantly delay the enteric release of butyric acid, thus reducing small intestinal absorption and enhancing colonic delivery [44,45,46]. This may explain the difference between our results and those reported in previous studies.

Numerous studies have revealed that the gut microbiota that improve growth and metabolism promote host nutrient absorption and modulate the immune system, which are all behaviors providing irreplaceable functionality [47,48]. Furthermore, the diversity of the microbial community helps to maintain the homeostasis of the intestinal microbiome and improve resistance to pathogens in the host [49]. Our results on alpha diversity and beta diversity in the cecum content revealed a degree of diversity discrepancy in the cecum microbiota. Both the *C. perfringens* challenge and addition of SB improved the diversity of bacterial flora. This result is not consistent with previous studies, such as that of Pammi et al. (2017), in which the intestinal flora diversity in necrotizing enterocolitis patients was lower than that in unaffected patients, such as the lower relative abundance of *Firmicutes* and *Bacteroides* and higher relative abundance of *Proteobacteria* [50]. Meanwhile, our results were consistent with those of Zhang et al. (2018), who revealed the α-diversity index of broiler gut microbial community after *C. perfringens* infection [51]. This may be explained by the fact that *C. perfringens* infection can destroy the ecological balance of intestinal flora, resulting in intestinal ecological imbalance [52]. In addition, *C. perfringens* strains and dietary components, as well as the timing and duration of the *C. perfringens* challenge, may have contributed to the observed discrepancy [53]. In this study, the results indicated that the proportions of *Bacteroides*, *Faecalibacterium*, *Clostridia*, *Ruminococcaceae*, *Alistipes,* and *Clostridia* were significantly higher in the MS-SB addition group; at the same time, *C. perfringens* infection increased the abundance of *Bacteroides* and *Alistipes*. In previous studies, the genera *Alistipes* and *Bacteroides* have been identified as butyrate producers in the gut and have demonstrated good anti-inflammatory effects through butyrate [54]. The results of the correlation analysis considering VFAs and microbiota indicated that the isobutyric acid content increased and was positively correlated with the abundance of *Alistipes* and *Clostridia*. Thus, our findings demonstrate that SB and MS-SB exhibit a protective role to suppress *C. perfringens*-induced intestinal damage and microbiota disturbances.

## 5. Conclusions

In summary, sodium butyrate ameliorated *C. perfringens* infection by reducing inflammation, repairing intestinal damage, and modulating the cecum microbiota. Compared to sodium butyrate, microencapsulated sodium butyrate presented a better effect. This study highlights the effectiveness of microencapsulated sodium butyrate and provides a novel strategy for protection against *C. perfringens* infection through the effective utilization of gastrointestinal-sensitive biological agents.

## Figures and Tables

**Figure 1 animals-13-03784-f001:**
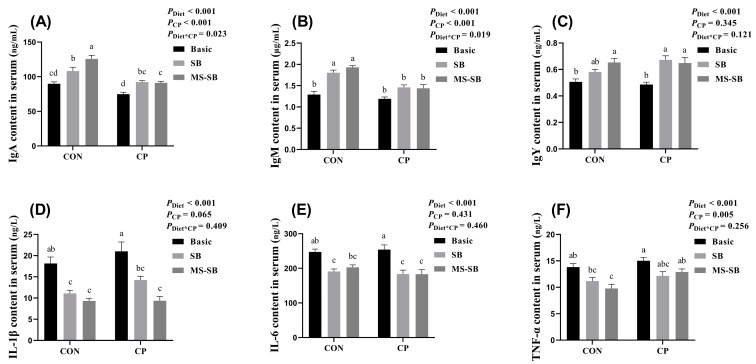
MS-SB modulates immunity through globulins and immune factor content in serum during *C. perfringens* infection. (**A**) IgA content; (**B**) IgM content; (**C**) IgY content; (**D**) IL-1β content; (**E**) IL-6 content; (**F**) TNF-α content. Bars with different letters are statistically significant (*p* ˂ 0.05) in different groups, *n* = 6.

**Figure 2 animals-13-03784-f002:**
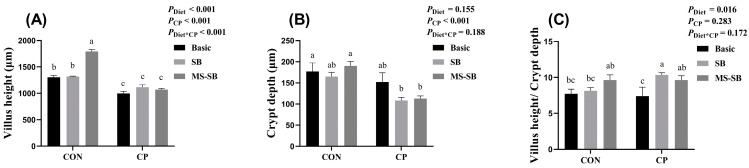
MS-SB repaired intestinal morphology damaged by *C. perfringens*. (**A**) Villus height of jejunum; (**B**) crypt depth of jejunum; (**C**) ratio of the villus height and crypt depth of jejunum. Bars with different letters are statistically significant (*p* ˂ 0.05) in different groups, *n* = 6.

**Figure 3 animals-13-03784-f003:**
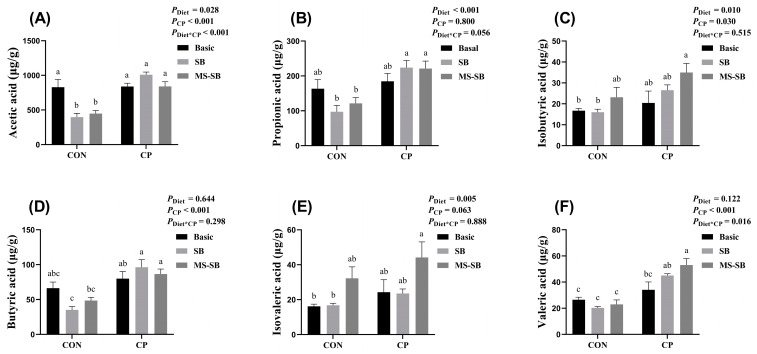
Microencapsulated sodium butyrate modulates the VFA content when challenged with *C. perfringens*. (**A**) Acetic acid; (**B**) propionic acid; (**C**) isobutyric acid; (**D**) butyric acid; (**E**) isovaleric acid; (**F**) valeric acid content in cecum of broiler chickens. Bars with different letters are statistically significant (*p* ˂ 0.05) in different groups, *n* = 6.

**Figure 4 animals-13-03784-f004:**
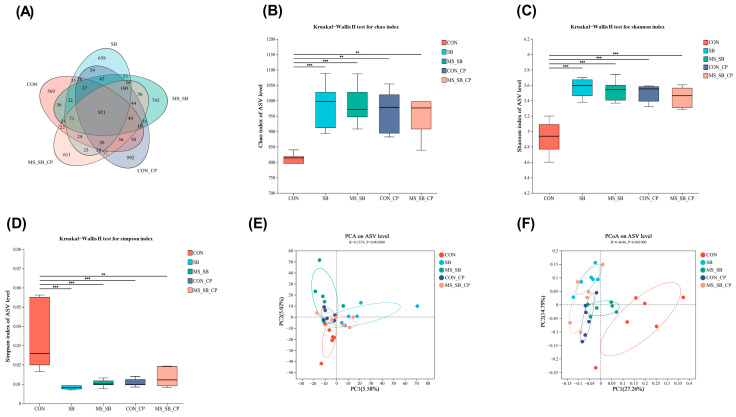
Analysis of the diversity of gut microbiota. (**A**) The Venn diagram summarizing the numbers of common and unique OTUs in cecum microflora community. (**B**–**D**) The Chao index, Shannon index, and Simpson index reflecting species alpha diversity between groups. (**E**,**F**) The principal component analysis (PCA) and principal co-ordinates analysis (PCoA) reflecting beta diversity within and between groups at the species level. The column with * are statistically significant, ** means *p* ˂ 0.01, *** means *p* ˂ 0.001, *n* = 6.

**Figure 5 animals-13-03784-f005:**
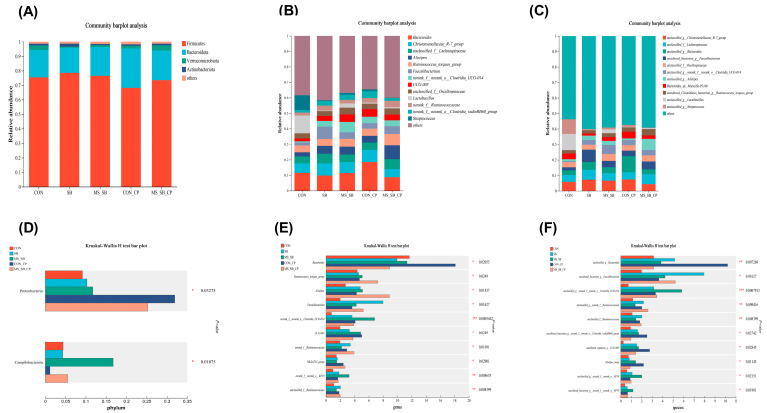
The abundance of the microbial community in cecum content. (**A**–**C**) The top relative abundance of the microflora community between groups (phylum level, genus level, and species level). (**D**–**F**) The bacteria with significant differences between groups (phylum level, genus level, and species level). The column with * are statistically significant, * means *p* ˂ 0.05, ** means *p* ˂ 0.01, *** means *p* ˂ 0.001, *n* = 6.

**Figure 6 animals-13-03784-f006:**
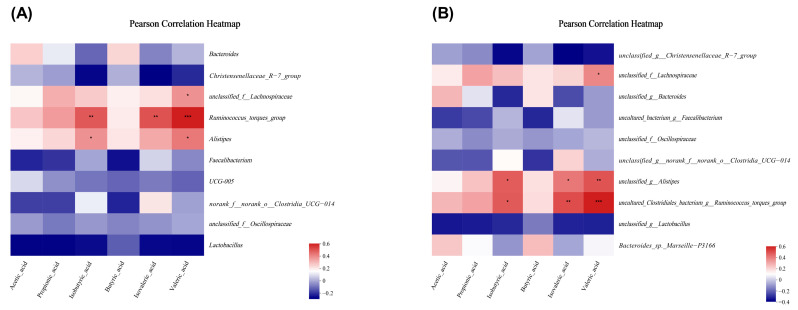
Correlation analysis between gut microbiota and VFAs. (**A**) Genus level and VFAs. (**B**) Species level and VFAs. The column with * are statistically significant, * means *p* ˂ 0.05, ** means *p* ˂ 0.01, *** means *p* ˂ 0.001, *n* = 6.

**Table 1 animals-13-03784-t001:** Ingredients and nutrient composition of base diets, as feed basis.

Ingredients (%)	Starter (Days 1–28)	Grower (Days 29–56)	Nutritional Level	Starter (Days 1–28)	Grower (Days 29–56)
Corn	54.4	53	Me (kcal/kg)	2983	3090
Soybean meal	23.6	16	CP (%)	20.4	17.2
Expanded soybean	5	3	Lysine (%)	1.18	0.96
Rice DDGS	5	8	Methionine (%)	0.55	0.44
Rice bran	/	8	Met + Cys (%)	0.90	0.74
Corn bran	/	2	Tryptophan (%)	0.22	0.20
Soybean oil	2.2	4.5	Threonine (%)	0.88	0.78
Limestone	1.5	1.9	Calcium (%)	0.86	0.73
Fermented soybean meal	2.5	/	Total P (%)	0.70	0.71
Corn gluten meal	2.0	/	Available P (%)	0.43	0.44
CaHPO_4_ (2H_2_O)	2.0	1.8			
NaCl	0.3	0.3			
Premix ^a^	1.5	1.5			
Total	100.00	100.00			

^a^ The following substances were supplied per kilogram of diet: vitamin A, 10,000 IU; vitamin D_3_, 2500 IU; vitamin E, 20 mg; vitamin B_1_, 1.5 mg; vitamin B_2_, 3.5 mg; pantothenic acid, 10 mg; vitamin B_12_, 0.01 mg; folic acid, 1 mg; niacin 30 mg; Choline chloride, 1000 mg; Cu (CuSO_4_·5H_2_O), 8 mg; Fe (FeSO_4_·7H_2_O), 80 mg; Zn (MnSO_4_·7H_2_O), 60 mg; Se (NaSeO_3_), 0.15 mg; I (KI), 0.2 mg.

**Table 2 animals-13-03784-t002:** Effect of microencapsulated sodium butyrate on the organ index of broiler chickens challenged with clostridium perfringens.

Diet	Challenge	Liver Index	Spleen Index	Bursa of Fabricius Index	Thymus Index
Control	CON	16.99	1.37	1.14	2.34 ^a^
SB		18.04	1.49	1.23	1.84 ^ab^
MS-SB		17.67	1.77	0.76	2.43 ^a^
Control	CP	17.77	1.84	0.53	1.46 ^bc^
SB		19.74	1.89	0.75	0.98 ^c^
MS-SB		17.25	1.62	0.92	0.97 ^c^
SEM		0.65	0.11	0.19	0.14
Main effect					
Diet	Control	17.38	1.60	0.83	1.90 ^a^
	SB	18.86	1.70	0.99	1.41 ^b^
	MS-SB	17.46	1.69	0.84	1.63 ^b^
Challenge	CON	17.56	1.54 ^b^	0.73	2.24 ^a^
	CP	18.25	1.78 ^a^	1.04	1.37 ^b^
*p*-value					
Diet		0.057	0.743	0.694	0.005
Challenge		0.221	0.034	0.063	<0.001
Diet × Challenge		0.304	0.051	0.128	0.023

Note. Number with different letters in the same column are statistically significant.

## Data Availability

The data that support the findings of this study were not deposited in an official repository, but they are available from the authors upon request.

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
