# Peer review of "Microencapsulated Sodium Butyrate Alleviates Immune Injury and Intestinal Problems Caused by Clostridium Perfringens through Gut Microbiota"

_animals, 2023, doi:10.3390/ani13243784_

Round 1

Reviewer 1 Report

Comments and Suggestions for Authors

Comments to the Authors of manuscript number: animals-2726404 entitled “Microencapsulated sodium butyrate alleviates immune injury and intestinal healthy caused by clostridium perfringens through gut microbiota”.

Microencapsulated sodium butyrate (MS-SB) and sodium butyrate (SB) were studied as additives for broilers challenged with Clostridium perfringens. Both improved immunity, intestinal health, and microbiota, with MS-SB showing better results.

Generally, the study design is very poorly describe as well as methodology. In this form the paper cannot be published.

1. L 4 – correct, check L 43

2. Microencapsulated sodium butyrate (MS-SB) and sodium butyrate (SB) were studied as additives for broilers challenged with Clostridium perfringens. Both improved immunity, intestinal health, and microbiota, with MS-SB showing better results.

3. L 41-42 – this sentence has no sense

4. L 46 – “ameliorated the mice”?

5. “As previous study, diet supplement of butyrate attenuated the inflammatory re- 68 sponse and improved intestinal health in piglets challenged with enterotoxigenic Esche- 69 richia coli (ETEC), which through inhibiting the NF-κB/MAPK activation and modulating 70 the hindgut microbiota” – not finished?

6. L 75- cite them

7. L 99-101- when was this experimental period? When birds were fed these diets? It is unclear

8. L 101- how long were beds fed these diet before challenge by C. perfringers?

9. where did the bacteria come from?

10. L 103- every day? How long?

11. L 103-"via" instead of "adopt."

12. The sentence starting with "The MS-SB used in this study" is incomplete or lacks clear context. It mentions the properties of MS-SB but doesn't connect well with the previous sentences. It could be improved for clarity and flow. What?

13. L 107-109 – It should be corrected

14. It is not clear why this chickens were chosen? Other chickens are kept until 42th day of age. It should be explained

15. L 109 – how many from group? How were chicken chosen? It should be explained

16.  L 124-128- catalog numbers of kits should be given and minimal detection

17. L 132- how many samples were done? How many data were finally? Everything should be presented.

18. L 136- how many samples?

19. L 172- correction needed

20. L 185, 186, 187- it is not content

21. Figure 1 should be corrected. It is not a content

22. What does it mean:” We also measured the morphology..”?? is it possible to measure morphology through staining???

23. MS-SB and SB are not described, the company should be included.

Author Response

Thanks to the reviewers  for your comments, which are very helpful to the improvement of this paper. For the questions or comments, we have made the following correction.

Question 1: 1.    L 4 – correct, check L 43

Thanks for your question. We have fixed similar problems throughout the article.

Question 2: 1. 4. L 46 – “ameliorated the mice”?

Thanks for your careful review, there should be “intestine” in here, we ignore this in the process of document processing.

Question 3: 5. “As previous study, diet supplement of butyrate attenuated the inflammatory re- 68 sponse and improved intestinal health in piglets challenged with enterotoxigenic Esche- 69 richia coli (ETEC), which through inhibiting the NF-κB/MAPK activation and modulating 70 the hindgut microbiota” – not finished?

Thanks for your careful review, the sentence was finished, the cite reference are the sign of the end.

Question 4: L 75- cite them.

Thanks for your careful review, the cite reference was [14] at line 78.

Question 5: the question 7-15 are about study design, we rewrite this part.

Thanks for your careful review, we have rewritten this section for greater clarity.

Question 6: L 124-128- catalog numbers of kits should be given and minimal detection.

Thanks for your careful review, we added the information about catalog numbers of kits.

Question 7: L 132- how many samples were done? How many data were finally? Everything should be presented.

           L 136- how many samples?

Thanks for your careful review, there are 6 samples, the number of samples are shown at figure legend.

Question 8: 19. L 172- correction needed

  1. L 185, 186, 187- it is not content

Thanks for your careful review, we have rewritten this sentence.

Question 9: 21. Figure 1 should be corrected. It is not a content

Thanks for your careful review, we notice the error in the figure 1, and re-exported new figure form graphpad.

Question10: 22. What does it mean:” We also measured the morphology.”?? is it possible to measure morphology through staining???

Thanks for your careful review, we agree with you very much, and changed the way of presentation.

Question10: 23. MS-SB and SB are not described, the company should be included.

Thanks for your careful review, we have modified it together with question 5.

Reviewer 2 Report

Comments and Suggestions for Authors

The manuscript is written on good theme. But the English langauge is written very poor. I suggest to revise language and then submit for proper review. Also please revise the title. It could be Microencapsulated sodium butyrate recovers immune injury and intestinal health induced by clostridium perfringens 3 through gut microbiota.....

Comments on the Quality of English Language

The English langauge is written very poor. I suggest to revise language and then submit for proper review. 

Author Response

Thanks to the reviewers  for your comments, which are very helpful to the improvement of this paper. For the questions or comments, we have made the following correction.

Question 1: This manuscript has been revised by an English editor.

Question 2: Also please revise the title. It could be Microencapsulated sodium butyrate recovers immune injury and intestinal health induced by clostridium perfringens 3 through gut microbiota.

Thanks for your suggestion, we think that the original title can better reflect the research content, after sorting out the content of the article.

Reviewer 3 Report

Comments and Suggestions for Authors

In this article, the author compared the protective effect and protection mechanism of microencapsulated sodium butyrate and sodium butyrate on Clostridium perus. Finally, it was found that adding microencapsulated sodium butyrate to the feed can better reduce inflammation, repair the intestine, and regulate the cecum microbiota to improve Clostridium gase infection. However, the readability of the manuscript can also be greatly improved. Through editing and some modifications, I think this manuscript will be more suitable for publication.

1.         Line 16, the broiler appearing in line 16 and the broiler chicken appearing in the keyword should be consistent.

2.         Line 22, V/C appears for the first time, and it is best to explain the full name.

3.         Line 26, there is a problem with the tense, “the results also show”, “show” should be the general past tense, changed to “showed”.

4.         The last sentence of the Abstract is a little abrupt, and MS-SB is not more effective than before. It is better to compare it before.

5.         Table 1, the ( % ) of the nutrition level in this column of Table 1 is best in the header, which is consistent with the first column.

6.         Table 2, the last line of the diet column, “diet*challenge” is not aligned in the middle, and the format is different from other lines.

7.         Figure 4 and Figure 5 are placed too densely. Multiple pictures are put together, and the content of the chart is not clear when viewed. It is recommended to put them separately in the attached table. The pictures should be concise and clear.

8.         A total of 59 references were cited for this article, but nearly 30 references were from before 2018 and 9 were from before 2010. For innovative articles, the reference time should be nearly  3-5 years. Please update the reference.

Comments on the Quality of English Language

Moderate editing of English language required

Author Response

Thanks to the reviewers  for your comments, which are very helpful to the improvement of this paper. For the questions or comments, we have made the following correction.

Question 1: 1.    1. Line 16, the broiler appearing in line 16 and the broiler chicken appearing in the keyword should be consistent.

  1. Line 22, V/C appears for the first time, and it is best to explain the full name.
  2. Line 26, there is a problem with the tense, “the results also show”, “show” should be the general past tense, changed to “showed”.

Thanks for your careful review, we agree with you very much, and correct those mistakes.

Question 2: 4. The last sentence of the Abstract is a little abrupt, and MS-SB is not more effective than before. It is better to compare it before.

Thanks for your careful review, we have rewritten this sentence.

Question 3: 5. Table 1, the (%) of the nutrition level in this column of Table 1 is best in the header, which is consistent with the first column.

  1. Table 2, the last line of the diet column, “diet*challenge” is not aligned in the middle, and the format is different from other lines.

Thanks for your careful review, we agree with you very much, and correct those mistakes.

Question 4: 7. Figure 4 and Figure 5 are placed too densely. Multiple pictures are put together, and the content of the chart is not clear when viewed. It is recommended to put them separately in the attached table. The pictures should be concise and clear.

Thanks for your careful review, we agree with you very much, and Improved picture quality through higher DPI. We will discuss with the editor and come up with a better solution for the layout of pictures.

Question 5: 8. A total of 59 references were cited for this article, but nearly 30 references were from before 2018 and 9 were from before 2010. For innovative articles, the reference time should be nearly 3-5 years. Please update the reference.

Thanks for your careful review, we agree with you very much, and update the reference as much as possible.

Round 2

Reviewer 1 Report

Comments and Suggestions for Authors

I have no comments

Reviewer 2 Report

Comments and Suggestions for Authors

Article has been improved significantly. However, I am still concerned with title "Intestinal healthy" word. Please check and may be corrected. 

The theme of the research is novel and manuscript is well written to present it with clarity.